# A ROTATION-EQUIVARIANT CONVOLUTIONAL NEURAL NETWORK MODEL OF PRIMARY VISUAL CORTEX

**Alexander S. Ecker,**[1-3,6,*] **Fabian H. Sinz,**[5,6] **Emmanouil Froudarakis,**[5,6] **Paul G. Fahey,**[5,6]
**Santiago A. Cadena**[1-3] **Edgar Y. Walker,**[5,6] **Erick Cobos,**[5,6]
**Jacob Reimer,**[5,6] **Andreas S. Tolias,** [1,5-7] **Matthias Bethge** [1-4,6]

[1] Centre for Integrative Neuroscience, University of Tübingen, Germany
[2] Bernstein Center for Computational Neuroscience, University of Tübingen, Germany
[3] Institute for Theoretical Physics, University of Tübingen, Germany
[4] Max Planck Institute for Biological Cybernetics, Tübingen, Germany
[5] Department of Neuroscience, Baylor College of Medicine, Houston, TX, USA
[6] Center for Neuroscience and Artificial Intelligence, BCM, Houston, TX, USA
[7] Department of Electrical and Computer Engineering, Rice University, Houston, TX, USA

[*] `alexander.ecker@uni-tuebingen.de`

## ABSTRACT

Classical models describe primary visual cortex (V1) as a filter bank of orientation-selective linear-nonlinear (LN) or energy models, but these models fail to predict neural responses to natural stimuli accurately. Recent work shows that convolutional neural networks (CNNs) can be trained to predict V1 activity more accurately, but it remains unclear which features are extracted by V1 neurons beyond orientation selectivity and phase invariance. Here we work towards systematically studying V1 computations by categorizing neurons into groups that perform similar computations. We present a framework for identifying common features independent of individual neurons' orientation selectivity by using a rotation-equivariant convolutional neural network, which automatically extracts every feature at multiple different orientations. We fit this rotation-equivariant CNN to responses of a population of 6000 neurons to natural images recorded in mouse primary visual cortex using two-photon imaging. We show that our rotation-equivariant network outperforms a regular CNN with the same number of feature maps and reveals a number of common features, which are shared by many V1 neurons and are pooled sparsely to predict neural activity. Our findings are a first step towards a powerful new tool to study the nonlinear functional organization of visual cortex.

## 1 INTRODUCTION

The mammalian retina processes image information using a number of distinct parallel channels consisting of functionally, anatomically, and transcriptomically defined distinct cell types. In the mouse, there are 14 types of bipolar cells (Euler et al., 2014), which provide input to 30–50 types of ganglion cells (Baden et al., 2016; Sanes & Masland, 2015). In visual cortex, in contrast, it is currently unknown whether excitatory neurons are similarly organized into functionally distinct cell types. A functional classification of V1 neurons would greatly facilitate understanding its computations just like it has for the retina, because we could focus our efforts on identifying the function of a small number of cell types instead of characterizing thousands of anonymous neurons.

Recent work proposed a framework for learning functional cell types from data in an unsupervised fashion while optimizing predictive performance of a model that employs a common feature space shared among many neurons (Klindt et al., 2017). The key insight in this work is that all neurons that perform the same computation but have their receptive fields at different locations, can be represented by a feature map in a convolutional network. Unfortunately, this approach cannot be applied directly to neocortical areas. Neurons in area V1 extract local oriented features such as edges at different

orientations, and most image features can appear at arbitrary orientations – just like they can appear at arbitrary locations. Thus, to define functional cell types in V1, we would like to treat orientation as a nuisance parameter (like receptive field location) and learn features independent of orientation.

In the present paper, we work towards this goal. While we do not answer the biological question whether there are indeed well-defined clusters of functional cell types in V1, we provide the technical foundation by extending the work of Klindt and colleagues (Klindt et al., 2017) and introducing a rotation-equivariant convolutional neural network model of V1. We train this model directly on the responses of 6000 mouse V1 neurons to learn a shared feature space, whose features are independent of orientation. We show that this model outperforms state-of-the-art CNNs for system identification and allows predicting V1 responses of thousands of neurons with only 16 learned features. Moreover, for most neurons, pooling from only a small number of features is sufficient for accurate predictions.

## 2 RELATED WORK

**Functional classification of cell types.** Characterizing neurons according to some identified, potentially nonlinear response properties has a long history. For instance, researchers have identified simple and complex cells (Hubel & Wiesel, 1968), pattern- and component-selective cells (Gizzi et al., 1990), end-stopping cells (Schiller et al., 1976), gain control mechanisms (Carandini et al., 1997) and many more. However, these properties have been identified using simple stimuli and the models that have been built for them do not apply to natural stimuli in a straightforward way. In addition, most of these studies were done using single-cell electrophysiology, which does not sample cells in an unbiased fashion. Thus, we currently do not know how important certain known features of V1 computation are under natural stimulation conditions and what fraction of neurons express them. Recent work using two-photon imaging in the monkey (Tang et al., 2018) started closing this gap by systematically investigating response properties of V1 neurons to an array of complex stimuli and found, for instance, that about half of the neurons in V1 are selective to complex patterns such as curvature, corners and junctions rather than just oriented line segments.

Recent work in the retina showed that different types of Ganglion cells (Baden et al., 2016) – and to some extend also bipolar cells (Franke et al., 2017) – can be identified based on functional properties. Thus, in the retina there is a relatively clear correspondence of anatomical and genetic cell types and their functional output, and we are getting closer to understanding each retinal cell type's function.

**Learning shared feature spaces for neural populations.** Antolík and colleagues pioneered the idea of learning a shared nonlinear feature representation for a large population of neurons (Antolík et al., 2016), which others have also used in both retina and V1 (Batty et al., 2017; Kindel et al., 2019; Cadena et al., 2019). Klindt et al. (2017) proposed a framework to learn functional cell types in an unsupervised fashion as a by-product of performing system identification. They propose a structurally constrained readout layer, which provides a compact characterization of each neuron's computation. By enforcing this representation to be sparse, they suggest that the learned features may correspond to distinct functional cell types. Our present work builds upon this idea and extends it to be applicable to V1.

**Rotation equivariance in CNNs.** There is a large body of work on equivariant representations (Nordberg & Granlund, 1996). Here we review only the most closely related approches. Cohen & Welling (2016) introduce group-equivariant CNNs, which include rotation equivariance as a special case and form the basis of our approach. Weiler et al. (2018) use a steerable filter basis to learn rotation-equivariant CNNs. We essentially use the same approach, but with a different set of steerable filters (2d Hermite functions instead of circular harmonics). RotEqNet (Marcos et al., 2017) is related to our work in the sense that it also applies multiple rotated versions of each filter to the input. However, instead of maintaining all resulting feature maps as inputs for the next layer, they apply an orientation pooling operation, which reduces each set of feature maps to a two-dimensional vector field. Harmonic networks (Worrall et al., 2017) are an alternative approach that achieves full $360°$ rotation equivariance by limiting the structure of the convolutional filters. Finally, Dieleman et al. (2016) achieve rotation equivariance by feeding multiple rotated versions of the input image into parallel CNN streams whose weights are tied together.

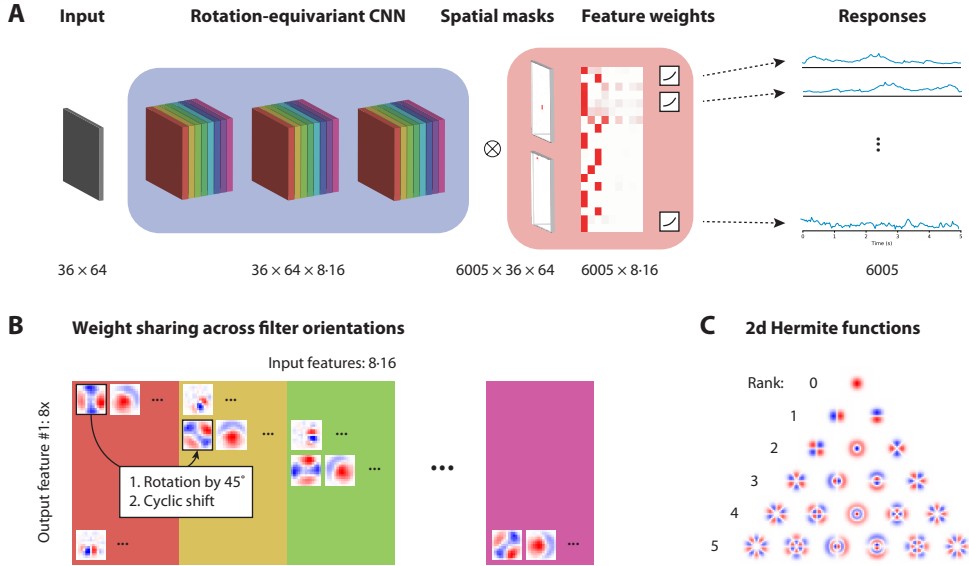

Figure 1: Rotation-equivariant CNN architecture. **A.** The network consists of a rotation-equivariant convolutional core common for all neurons (blue box) and a neuron-specific readout [red box (Klindt et al., 2017)]. Inputs are static images from ImageNet. Prediction targets are responses of 6005 V1 neurons to these images. Rotation equivariance is achieved by using weight sharing across filter orientations. Therefore, eight rotated versions of each filter exist, resulting in eight groups of feature maps (depicted by rainbow colored boxes). **B.** Illustration of weight sharing across orientations for the second and third layer. The previous layer's output consists of eight groups of feature maps (one for each rotation angle). The output is generated by applying a learned filter to each of the input feature maps (here $8 \times 16$ kernels shown in the first row). Rotated versions (2nd and following rows) are generated by rotating each kernel and cyclically permuting the feature maps. **C.** Filters are represented in a steerable basis (2d Hermite functions). Functions up to rank 5 are shown here.

## 3   ROTATION-EQUIVARIANT CNN

**Network architecture.**   Our architecture follows that of Klindt et al. (2017). The image is first processed by a multi-layer convolutional core shared by all neurons (Fig. 1A, blue), which we describe in detail below. The resulting feature representation is then turned into a response prediction for each neuron by applying a sparse linear combination across space, followed by a sparse linear combination across features and a pointwise, soft-thresholding nonlinearity (Fig. 1A, red). The entire model is trained end-to-end directly on predicting neural responses, without employing transfer learning or pre-training on auxiliary tasks (Yamins et al., 2014; Cadena et al., 2019).

This model has a number of desirable properties in terms of data efficiency and interpretability. The separation into convolutional core and readout, together with the strong structural constraints on the readout pushes all the 'heavy lifting' into the core while the readout weights (spatial mask and feature weights) provide a relatively low-dimensional characterization of each neuron's function. Because the core is shared among all neurons (here: thousands), many of which implement similar functions, we can learn complex non-linear functions very accurately.

**Equivariance.**   A function $f : \mathcal{X} \to \mathcal{Y}$ is called equivariant with respect to a group of transformations $\Pi$ if such transformations of the input lead to predictable transformations of the output. Formally, for every $\pi \in \Pi$ there is a transformation $\psi \in \Psi$ such that for every $x \in \mathcal{X}$

$$\psi[f(x)] = f[\pi(x)]. \tag{1}$$

CNNs are shift-equivariant by construction, meaning that every translation of the image leads to a matching translation in the feature maps (i. e. $\pi$ and $\psi$ are both translation by a number of pixels horizontally and vertically). Shift equivariance is a useful property for neural system identification,

because it allows us to represent many neurons that perform similar computations but in different locations in space by a single convolutional feature map instead of learning each neurons' nonlinear input-output function 'from scratch.'

**Rotation-equivariant CNN.** Neurons in V1 do not only perform similar functions at different locations, but also extract similar features with different orientations. Thus, for modeling populations of V1 neurons, learning a representation that is equivariant to rotation in addition to translation would be desirable. To achieve rotation equivariance, we use group convolutions (Cohen & Welling, 2016). Here we use the group of all rotations by multiples of $45°$. That is, for each convolutional filter in the first layer, we have eight rotated copies, each of which produces one feature map (Fig. 1A, blue). Thus, if we learn 16 different filters in the first layer, it will have a total of $8 \times 16 = 128$ feature maps. Formally, if $\pi$ is a rotation of the image by, say, $45°$, then $\psi$ is a rotation of the feature maps by also $45°$ combined with a cyclic permutation of the feature maps by one rotation step.

For the second (and all subsequent) layers, the procedure becomes a bit more involved. Sticking with the numbers from above, for every feature in the second layer we now learn 128 filters – one for each input feature map (Fig. 1B, first row). To preserve rotation equivariance, we need to create all rotated copies of these filters and cyclically permute the feature maps such that each rotated filter receives the rotated version of the input (depicted by the colored boxed in Fig. 1B, second and following rows).

To implement weight sharing across filter orientation without aliasing artifacts, we represent the filters in a steerable basis (Weiler et al., 2018). We use the two-dimensional Hermite functions in polar coordinates, which form a steerable, orthonormal basis (Fig. 1C; see also Victor et al., 2006; Hu & Victor, 2016). For filters of size $k$, we use all 2d Hermite functions up to rank $k$, which means we have $k(k + 1)/2$ basis functions. We sample each filter at twice the resolution and then downsample by $2 \times 2$ average pooling to reduce aliasing.

## 4 EXPERIMENTS

**Neural data.** We recorded the responses of 6005 excitatory neurons in primary visual cortex (layers 2/3 and 4) from one mouse by taking two consecutive scans with a large-field-of-view two-photon mesoscope (Sofroniew et al., 2016). Activity was measured using the genetically encoded calcium indicator GCaMP6f. V1 was targeted based on anatomical location as verified by numerous previous experiments performing retinotopic mapping using intrinsic imaging. We selected cells based on a classifier for somata on the segmented cell masks and deconvolved their fluorescence traces (Pnevmatikakis et al., 2016). We did not filter cells according to visual responsiveness. The aquisition frame rate was 4.8 Hz in both scans. We monitored pupil position, dilation, and absolute running speed of the animal. However, because eye movements were rare and we are primarily interested in the average response of neurons given the visual stimulus, we did not further take into account eye position and running speed.

**Visual stimuli.** Stimuli consisted of 5500 images taken from ImageNet (Russakovsky et al., 2015), cropped to fit a 16:9 monitor, and converted to gray-scale. The screen was $55 \times 31$ cm at a distance of 15 cm, covering roughly $120° \times 90°$. In each scan, we showed 5400 of these images once (training and validation set) and the remaining 100 images 20 times each (test set). Each image was presented for 500ms followed by a blank screen lasting between 500ms and 1s. For each neuron, we extract the accumulated activity between 50ms and 550ms after stimulus onset using a Hamming window.

**Preprocessing.** We rescaled the images to $64 \times 36$ pixels and standardized them by subtracting the mean over all images in the training set and all pixels from each image and dividing by the standard deviation (also taken over all images in the training set and all pixels). We divided the responses of each neuron by its standard deviation over time. We did not center the neural responses, because they are non-negative after deconvolution and zero has a clear meaning.

**Model fitting and evaluation.** We initialized all weights randomly from a truncated normal distribution with mean zero and standard deviation 0.01. The biases of the batch normalization layers were initially set to zero. In contrast, we set the biases in each neuron's readout to a non-zero initial value, since the neural responses are not centered on zero. We initialized these biases such that the initial model prediction was on average half the average response of each neuron. To fit the models, we used

the Adam Optimizer (Kingma & Ba, 2016) with an initial learning rate of 0.002, a single learning rate decay and early stopping. We monitored the validation loss every 50 iterations and decreased the learning rate once by a factor of 10 when the validation loss had not decreased for five validation steps in a row. We then further optimized the model until the same criterion is reached again.

To evaluate the models, we use the following procedure. For each neuron we compute the Pearson correlation coefficient between the model prediction and the average response over the 20 repetitions of the 100 test images. We then average the correlations over all neurons. This approach tells us how well the model predicts the average response of neurons to a given stimulus, ignoring trial-to-trial variability (which is interesting in itself, but not the focus of the present work).

**Architecture details.** Our architecture consists of three convolutional layers with filter sizes of 13, 5 and 5 pixels, respectively. Thus, the receptive fields of the CNN's last layer's units were 21 px, corresponding to $\sim 60°$ and covering both classical and extra-classical receptive field. We use zero padding such that the feature maps maintain the same size across layers. We use 16 filter sets (i. e. 128 feature maps) in the first two layers and 8 to 48 filter sets in the third layer (number cross-validated, see below). After each layer, we apply batch normalization followed by a learned scale and bias term for each feature map.[1] After the first and second layer, but not after the third, we apply a soft-thresholding nonlinearity $f(x) = \log(1 + \exp(x))$. The feature maps of the third layer then provide the input for each neuron's readout, which consists of a linear combination first over space and then over features, followed by an added bias term and a final soft-thresholding nonlinearity. Thus, each neuron implements a cascade of three LN operations.

**Regularization.** We use the same three regularizers as (Klindt et al., 2017). For smoothness of convolution kernels we set the relative weight of the first layer to twice as strong as in the second and third layer to account for the larger kernels. We apply group sparsity to the convolution kernels of the second and third layer. We regularize the spatial masks and feature weights in the readout to be sparse by applying the L1 penalty on the 3d tensor that results from taking their outer tensor product. The weights of all three regularizers are cross-validated as described in the next paragraph.

**Model selection.** We cross-validated over the number of filter sets in the third layer, ranging from 8 to 48 (i. e. 64 to 384 feature maps[2]) and the strength of the regularizers. For each architecture, we fit 32 models with different initial conditions and randomly drawn hyperparameters (smoothness of filters: 0.001–0.03, group sparsity of filters: 0.001–0.1, sparsity of readout: 0.005–0.03) and chose the best one according to its loss on the validation set (i. e. not using the test set). For all baseline and control models (see below), we also cross-validated over 32 randomly sampled sets of hyperparameters drawn from the same range of values.

**Baseline and control experiments.** As a baseline, we fit a number of regular CNNs without rotation equivariance. These models are completely identical in terms of number of layers, filter sizes, readout layer and fitting procedure, except for the weight sharing constraint across orientation. We fit models with the same number of feature maps as their rotation-equivariant counterparts as well as smaller models with fewer feature maps (see Table 1).

Previous work has enforced the feature weights in the readout to be positive (Klindt et al., 2017). Because we do not enforce such constraint in the present work, we ran a control experiment, in which we enforce the readout weights (both masks and feature weights) to be positive.

Sparsity of the spatial masks is well justified, because we know that receptive fields are localized. However, it is unclear whether sparsity on the feature weights is desirable, since the function each neuron implements may not be well aligned with the coordinate axes spanned by the small number of features we learn. Thus, we also fit a model without the sparsity regularizer for the feature weights.

---

[1]In our experiments the network did not implement exact rotation equivariance, because batch normalization was applied to each feature map individually instead of jointly to all rotated versions. We therefore re-ran a subset of experiments where we corrected this issue and verified that model performance was indistinguishable.

[2]384 feature maps was the maximum we could fit into 16 GB of available GPU memory.

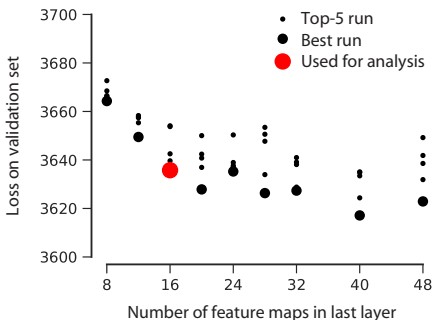

| Model | Corr. | S.D. |
|---|---|---|
| **Rotation-equivariant CNN** | | |
| **— 16x8–16x8–16x8** | **0.47** | **0.005** |
| — positive feature weights | 0.43 | 0.006 |
| — no feature sparsity | 0.43 | 0.02 |
| CNN (Klindt et al., 2017) | | |
| — 32–32–32 | 0.39 | 0.009 |
| — 64–64–64 | 0.39 | 0.006 |
| — 128–128–128 | 0.42 | 0.008 |
| — 128–128–256 | 0.41 | 0.009 |

Figure 2: Model comparison via loss on the validation set. Large dots: model with lowest loss on validation set; small dots: top five model fits according to loss on validation set. Red dot: model used for subsequent analyses. 16 feature maps provides a good trade-off between model complexity and predictive performance.

Table 1: Average correlation on test set of our rotation-equivariant CNN, two controls, and several regular CNNs as baselines. The three numbers for the CNNs are the number of feature maps in each layer; other parameters are identical to the rotation-equivariant CNN. Standard deviations are across the top-five models of each type.

**Tools.** We performed our model fitting and analyses using DataJoint (Yatsenko et al., 2015), Numpy/Scipy (Walt et al., 2011), Matplotlib (Hunter, 2007), Seaborn (Waskom et al., 2017), Jupyter (Kluyver et al., 2016), Tensorflow (Abadi et al., 2015), and Docker (Merkel, 2014).

**Availability of code and models.** Code to reproduce all experiments and models as well as pretrained models are available at `https://github.com/aecker/cnn-sys-ident`.

## 5 RESULTS

**Architecture search and model selection.** We start by evaluating the predictive performance of our rotation-equivariant CNN on a dataset of 6005 neurons from mouse primary visual cortex. First, we performed an initial architecture search, drawing inspiration from earlier work (Klindt et al., 2017) and exploring different numbers of layers and feature maps and different filter sizes. We settled on an architecture with three convolutional layers with filter sizes of 13, 5 and 5 pixels, respectively, and used 16 filter sets (i. e. 128 feature maps) in the first two layers. For the third layer, which provides the nonlinear feature space from which all neurons pool linearly, we cross-validated over the number of features, ranging from 8 to 48 (i. e. 64 to 384 feature maps).

Our model achieved an average correlation on the test set of 0.47. The performance improved with the number of features in the third layer, but the improvement was small beyond 16 features (Fig. 2). For the following analyses we therefore focus on this model with 16 features as a compromise between model complexity and performance.

It is encouraging that a model with only 16 features is sufficient to accurately model a population as large as 6000 neurons. Earlier work modeling V1 responses used a shared feature space of 10–20 (Antolík et al., 2016) or 48 features (Klindt et al., 2017) for populations of 50–100 neurons, which reduced the dimensionality of the feature space by a factor of 2–5 compared to the number of neurons. Here, we reduce the dimensionality by a factor of 375 while achieving a similar level of performance.

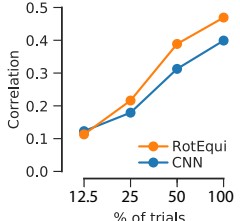

Figure 3: Performance vs. training data.

**Rotation-equivariant CNN outperforms regular CNN.** To compare our new model to existing approaches, we also fit a regular CNN with identical architecture except for the weight sharing constraint across orientations. In addition, we fit a number of smaller CNNs with fewer feature maps in each layer, which are more similar in terms of number of parameters, but potentially have less expressive power. Our rotation-equivariant CNN outperforms all baselines (Table 1) and generally requires less data (Fig. 3), showing that enforcing weight sharing across orientations is not only potentially useful for

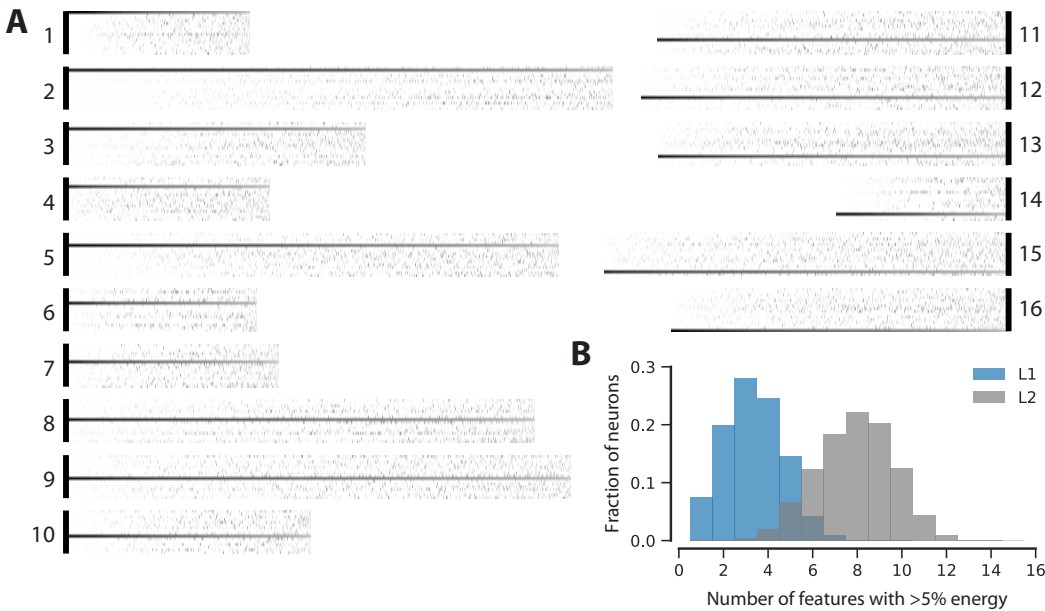

Figure 4: Feature weights are sparse. **A.** We group the neurons into 16 groups by their strongest feature weight. Within each group, we sort the neurons by the sparsity of their weights (in descending order). Within each block labeled 1–16, each row is one feature and each column is one neuron. **B.** Distribution of 'active' features over neurons (blue: $L_1$ regularization; gray: $L_2$ control).

interpreting the model (as we show below), but also serves as a good regularizer to fit a larger, more expressive model.

**Feature space generalizes to unseen neurons.** To show that our network learns common features of V1 neurons, we excluded half of the neurons when fitting the network. We then fixed the rotation-equivariant convolutional core and trained only the readout (spatial mask and feature weights) for the other half of the neurons. The resulting test correlation for these neurons (0.46) was indistinguishable from that of the full model (0.47), showing that the learned features transfer to neurons not used to train the feature space.

**Feature weights are sparse.** The intuition behind the sparse, factorized readout layer is that the spatial mask encodes the receptive field location of each neuron while the feature weights parameterize the neuron's nonlinear computation. We now ask how the neurons are organized within this 16-dimensional function space. On the one extreme of the spectrum, each neuron could pick a random direction, in which case sparsity would not be the right prior and the feature weights should be dense. On the other end of the spectrum, there could be a discrete number of functional cell types. In this case, each cell type would be represented by a single feature and the feature weights should be maximally sparse, i. e. one-hot encodings of the cell type.

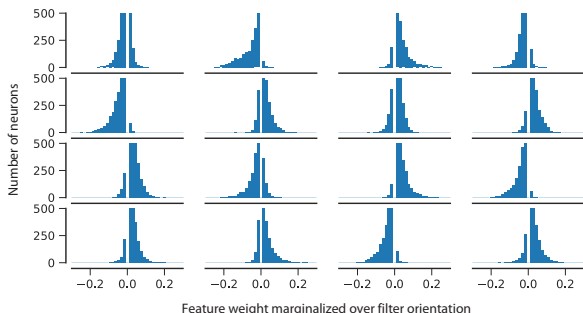

Figure 5: Feature weights have consistent sign across neurons. Note that the majority of weights is zero; the central bin has been excluded for clarity.

To analyze the feature weights, we first marginalize them over orientation. To do so, we take the sum of squares over all 8 orientations for each of the 16 features and then normalize them such that the energy of the 16 features sums to one. We find that the feature weights are indeed quite sparse

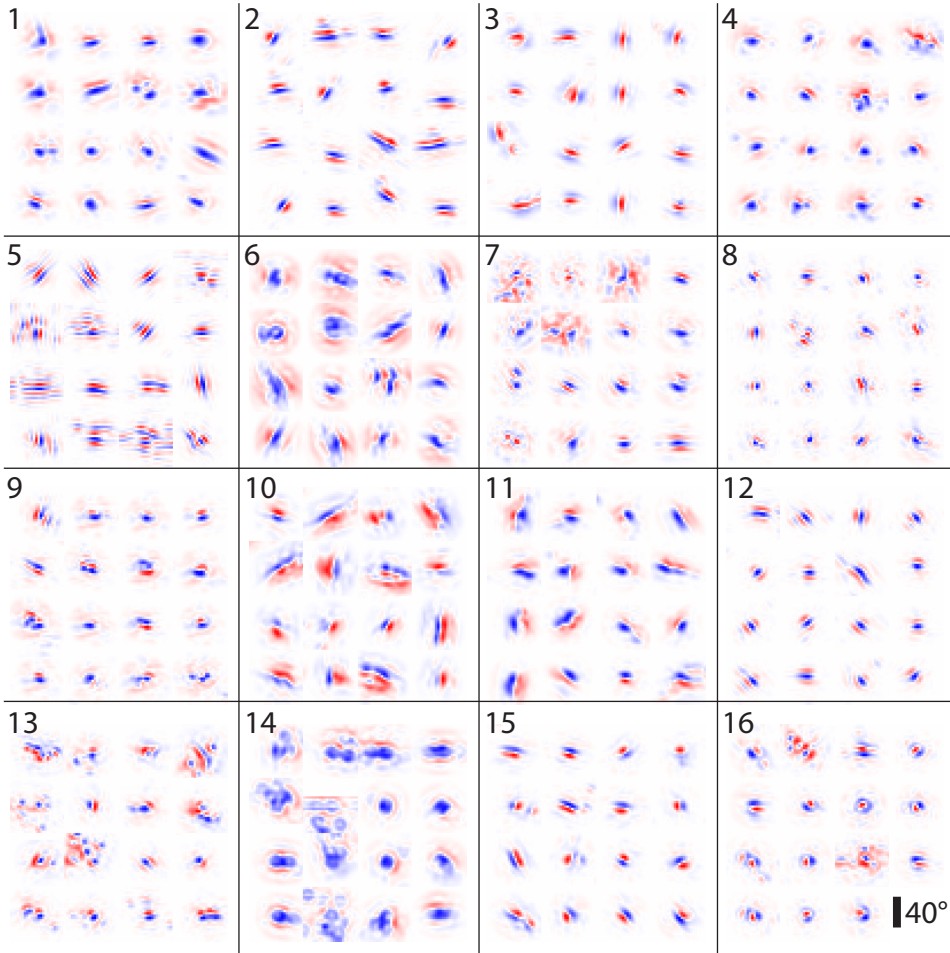

Figure 6: Linear receptive fields for all 16 groups of neurons, obtained by computing the gradient of the model at a constant gray image. For each group, we show the 16 neurons with the sparsest feature weights that have an average correlation of the model prediction with the neural response of at least 0.2 on the validation set. Thus, neurons are in the upper $50^{\text{th}}$ percentile, but not cherry-picked to be particularly well-predicted examples. Crops are roughly $70 \times 70°$.

(Fig. 4A): most of the neurons use only 1–5 features (Fig. 4B) and the strongest feature captures more than 50% of the energy for 63% of the neurons.

To ensure that the sparsity of the feature weights is a property of the data and not a trivial result of our $L_1$ penalty, we performed an ablation study, where we fit a model that applies sparsity only on the spatial masks, but uses $L_2$ regularization for the feature weights. This model performed worse than the original model (Table 1) and produced a significantly denser weight distribution (Fig. 4B), suggesting that sparsity is indeed a justified assumption.

**Feature weights have consistent sign.** There is a second line of evidence that the features learned by our model are meaningful beyond just providing an arbitrary basis of a 16-dimensional feature space in which neurons are placed at random: the weights that different neurons assign to any given feature have remarkably consistent signs (Fig. 5). For 11 of 16 features, more than 90% of the non-zero weights have the same sign.[3] Thus, the negation of one feature that drives a large group of neurons is not a feature that drives many neurons.

---

[3]Note that the sign itself does not carry any meaning. Because there is no nonlinearity after the last convolution, we can flip the sign of all filters generating this feature and at the same time flip all neurons' readout weights for this feature, which would leave the model prediction unchanged.

**Visualization of the different groups of neurons.**   Having presented evidence that the features identified by our model do indeed represent a simple and compact, yet accurate description of a large population of neurons, we now ask what these features are. As a first step, we group all neurons into one of 16 groups based on their strongest feature weight. This approach is obviously to some extent arbitrary. By no means to we want to argue that there are exactly 16 types of excitatory neurons in V1 or that excitatory neurons in V1 can even be classified into distinct functional types. Nevertheless, we believe that the 16 groups defined in this way are useful in a practical sense, because they represent features that cover more than 50% of the variance of a large fraction of neurons and therefore yield a very compact description of the most important features of V1 computation across the entire excitatory neuronal population.

To visualize what each of these features compute, we pick the 16 most representative examples[4] of each group and use the model to approximate their linear receptive field (RF). To this end, we compute the model gradient at a gray image (Fig. 6). The crucial point of this plot is that it shows that the premise of rotation equivariance holds: cells are clustered according to similar spatial patterns, but independent of their preferred orientation. As expected, most linear RFs resemble Gabor filters, which show differences in symmetry (odd: #2, #11, #15 vs even: #3, #6, #12) as well polarity (#3 vs. #6, #12), while some groups exhibit center-surround structure (#4, #14, #16).

## 6   DISCUSSION

We developed a rotation-equivariant convolutional neural network model of V1 that allows us to characterize and study V1 computation independent of orientation preference. Although the visual system is not equivariant to rotation – there are known biases in the distribution of preferred orientations –, enforcing weight sharing across orientations allowed us to fit larger, more expressive models given a limited dataset. While our work lays out the technical foundation, we only scratched the surface of the many biological questions that can now be addressed. Future work will have to investigate the learned features in much more detail, test to what extent they generalize across recording sessions and animals, whether they are consistent across changes in the architecture and – most importantly – whether neurons in V1 indeed cluster into distinct, well-defined functional types and this organization finds any resemblance in anatomical or genetic properties (Tasic et al., 2018) of the neurons recorded.

## ACKNOWLEDGMENTS

Supported by the Deutsche Forschungsgemeinschaft (DFG, German Research Foundation) via grant EC 479/1-1 to A.S.E and the Collaborative Research Center (Projektnummer 276693517 – SFB 1233: Robust Vision); the German Federal Ministry of Education and Research through the Tübingen AI Center (FKZ 01IS18039A); the International Max Planck Research School for Intelligent Systems (IMPRS-IS) supporting S.A.C; the Bernstein Center for Computational Neuroscience Tübingen; the Intelligence Advanced Research Projects Activity (IARPA) via Department of Interior/Interior Business Center (DoI/IBC) contract number D16PC00003. The U.S. Government is authorized to reproduce and distribute reprints for Governmental purposes notwithstanding any copyright annotation thereon. Disclaimer: The views and conclusions contained herein are those of the authors and should not be interpreted as necessarily representing the official policies or endorsements, either expressed or implied, of the National Institutes of Health, IARPA, DoI/IBC, or the U.S. Government.

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
