# OpenReview forum: "A rotation-equivariant convolutional neural network model of primary visual cortex"
_ICLR.cc/2019/Conference_

### Official Review · AnonReviewer2 · 2018-10-17
**Interesting work on matching CNN filters to Neurons**

**Rating:** 8
**Confidence:** 3

**Review:**

The paper analyses the data collected from 6005 neurons in a mouse brain. Visual stimuli are presented and the responses of the neurons recorded. In the next step, a rotational equivariant neural network architecture together with a sparse coding read-out layer is trained to predict the neuron responses from the stimuli. Results show a decent correlation between neuron responses and trained network. Moreover, the rotational equivariant architecture beats a standard CNN with similar number of feature maps. The analysis and discussion of the results is interesting. Overall, the methodological approach is good.

I have trouble understanding the plot in Figure 4, it also does not print well and is barely readable on paper.

I have a small problem Figure 6 where "optimal" response-maps are presented. From my understanding, many of those feature maps are not looking similar to feature maps that are usually considered. Given the limited data available and the non-perfect modeling of neurons, the computed optimal response-map might include features that are not present in the dataset. Therefore, it would be interesting to compare those results with the stimuli used to gather the data. E.g. for a subset of neurons, one could pick the stimulus that created the maximum response and compare that to what the stimulus with the maximum response of the trained neuron was. It might be useful to include the average correlation of the neurons belong to each of the 16 groups(if there are any meaningful differences), especially as the cut-off of "correlation 0.2 on the validation set" is rather low.

Note: I am not an expert in the neural-computation literature, I am adapting the confidence rating accordingly.

---

> ### Author Response · Authors · 2018-11-13
> **We will address the comments about Fig. 6 with additional analyses**
>
> Thank you for reviewing our paper and providing constructive comments.
>
> Regarding the optimal stimuli presented in Fig. 6: Thank you for this suggestion. We will investigate whether an analysis of the optimal stimulus in the training set is helpful. Unfortunately, this may not be the case, because neural responses are highly variable and therefore the stimulus that happened to elicit the strongest response may not actually be the one the drives the neuron most on average. In addition, we have done such analysis in the past on networks trained for large-scale object recognition and found that one needs several hundred thousand image patches to find examples close to the optimum (and these networks do not have observation noise). Note, however, that the variability across neurons within one group is a good indicator of which aspects of the optimal stimuli are reproducible vs. noise, because the neurons have different receptive field locations and have therefore seen different stimuli during the experiment. See also the suggestion by AnonReviewer1 to split the data in two halves, which we will perform for the final version.
>
> We will add the average correlations of the neurons shown in Fig. 6 as you suggested.
>
> Regarding your trouble with Fig. 4: We will make it full width, so it is more legible. Overall, the point of the figure is to show that for a large fraction of neurons the weights are very sparse. We are open to any suggestions on how to improve the Figure.

---

### Official Review · AnonReviewer1 · 2018-10-29
**Does rotation equivariance add neuroscientific insight?**

**Rating:** 5
**Confidence:** 4

**Review:**

﻿This paper applies a rotation-equivariant convolutional neural network model to a dataset of neural responses from mouse primary visual cortex. This submission follows a series of recent papers using deep convolutional neural networks to model visual responses, either in the retina (Batty et al., 2016; McIntosh et al., 2016) or V1 (Cadena et al., 2017; Kindel et al., 2017; Klindt et al., 2017). The authors show that adding rotation equivariance improves the explanatory power of the model compared to non-rotation-equivariant models with similar numbers of parameters, but the performance is not better than other CNN-based models (e.g. Klindt et al., 2017). The main potential contributions of the paper are therefore the neuroscientific insights obtained from the model. However, I have concerns about the presented data and the validity of rotation equivariance in modeling visual responses in general (below). Together with the fact that the model does not provide better explanatory power than other models, I cannot recommend acceptance. I am open to discussions with the authors, but do not anticipate a major change in the rating.

Update after revisions: The authors performed extensive work to address my concerns. This showed that some concerns (RF appearance) were valid, and the authors removed them from the final manuscript. I raised my score accordingly.

1. As noted by the authors, the finding that “Feature weights are sparse” (page 6) could be due to the sparsity-inducing L1 penalty. The fact that a model without L1 penalty performs worse does not mean that there is sparsity in the underlying data. For example, the unregularized model could be overfitting. A more careful model selection analysis is necessary to show that the data is better fit by a sparse than a dense model.

2. The finding that there are center-surround or asymmetric (non-gabor) RFs in mouse V1 is not novel and not specific to this model (e.g. Antolik et al., 2016).

3. Many of the receptive fields in Figure 6 look pathological (overfitted?) compared to typical V1 receptive fields in the literature. I understand that sensitivity to previously undetected RF features is a goal of the present work. However, given how unusual the RFs look, more controls are necessary to ensure they are not an artefact of the method, e.g. the activation maximization approach with gradient preconditioning, the sparsity constraints, or overfitting. Perhaps a comparison of RFs learned on two disjoint subsets of the training set would help to determine which features are reproducible.

4. Should orientation be treated as a nuisance variable? Natural image statistics are not rotation-invariant. In the visual system, especially in mice, it is not clear whether orientation is completely disentangled from other RF properties. The orientation space is not uniformly covered, and some directions have special meaning (e.g. cardinal directions), such that it might be invalid to assume that the visual system is equivariant to rotation. (The same concern applies to the translation equivariance assumed when modeling visual RFs with standard CNNs.) Of course, there is a tradeoff between model expressiveness and the need to make assumptions to fit the model with realistic amounts of data. However, this concern should at least be discussed.

5. Some more details about the neural recordings would be good. What calcium indicator? How was the recording targeted to V1? Perhaps some example traces.

---

> ### Author Response · Authors · 2018-11-06
> **Rotation-equivariant model does beat regular CNN**
>
> Thank you for reviewing our paper. We would like to make a quick clarification right away, which we hope will change your assessment. We will provide a more detailed response to the other comments later.
>
> There seems to be a misunderstanding about the performance of the model. As shown in Table 1, our rotation-equivariant CNN does outperform a regular CNN (Klindt et al. 2017).
>
> A couple of more detailed points to also keep in mind in this respect:
> - We are quite conservative with the model comparison: Table 1 shows the rotation-equivariant model with 16 features, which is not even the best-performing one among all the rotation-equivariant ones we tested (Fig. 2).
> - Related to above, the regular CNN has been subjected to an equally rigorous hyperparameter search, with the range of hyperparameters taken from Klindt et al (2017). Thus, the comparison is as fair as we can make it.
> - The performance in absolute numbers is lower than that in Klindt et al. (2017), but these numbers are not comparable because different datasets are used. There is quite some variability across datasets (see, e.g., Table 1 in Klindt et al. 2017).

---

> > ### Comment · AnonReviewer1 · 2018-11-06
> > **Performance was comparable to earlier studies**
> >
> > Thanks for the comment. A fair assessment, given the variability between datasets, would be that your model performs similarly to previous studies, as you stated in the text ("Performance was comparable to earlier studies modeling V1 responses with similar stimuli (Klindt et al., 2017; Antolík et al., 2016).
> >
> > I agree that you made a very rigorous effort in comparing the models (e.g. the hyperparameter search for control models).
> >
> > My thinking was that an improvement in model fit quality cannot be considered a main contribution of the paper, because the difference is within measuring variability. In any case, I don't think it is essential to show a performance improvement. The insights gained from the model are more important. This is what my main comments are about.

---

> > > ### Author Response · Authors · 2018-11-06
> > > **One should not compare performance across studies**
> > >
> > > Thanks for your quick reply and the clarification. We will address the other points shortly. Just one more quick comment about performance, because we think it's important to be on the same page.
> > >
> > > For a given dataset, the difference in performance between our model and a conventional CNN is *not* within measuring variability. We report the SD of the best models in Table 1, and the difference is several SDs between the two models.
> > >
> > > The difference in performance *across datasets* is not very meaningful, because the absolute numbers depend on factors such as the signal-to-noise ratio of the recordings, the brain state of the animal and the number of repeats per image. We should have avoided that one sentence about performance being comparable to earlier studies for this very reason.

---

> ### Author Response · Authors · 2018-11-13
> **Careful review, but the score seems too harsh or based in part on misunderstandings**
>
> Thank you for your careful review. We would like to start by clarifying the contributions of our paper before providing responses to your five specific points.
>
> As also mentioned in the response to AnonReviewer3, our long-term goal is to find out whether V1 is organized in distinct, well-defined clusters of functional cell types. However, this is a complex biological question that will require additional, very careful and extensive data analysis as well as potentially further direct experimental verification. The contribution of the present paper are therefore not the biological insights, but instead the development and verification of methods that will allow us to address this question. Specifically, this means (1) adapting rotation-equivariant CNNs to the problem of predicting neural responses, (2) showing that it can successfully be trained using a steerable basis for the filters, (3) showing that it outperforms conventional CNNs, and (4) showing that it does so with substantially fewer features than previous methods. We revised the introduction to make this point more explicit.
>
> In addition, on re-reading your comments, we believe there may be another misunderstanding regarding performance that we would like to clarify. You write:
>
> “The authors show that adding rotation equivariance improves the explanatory power of the model compared to non-rotation-equivariant models with similar numbers of parameters, but the performance is not better than other CNN-based models (e.g. Klindt et al., 2017).”
>
> This sentence mentions “non-rotation-equivariant models” and “other CNN-based models (e.g. Klindt et al., 2017)” as if they were two different entities. However, the non-rotation-equivariant model we use is exactly that of Klindt et al. 2017 (more or less literally, their code is public). So our model *does perform better* than that of Klindt et al. 2017. The only reason why the numbers are not better is because it is a different (larger) dataset.
>
> We hope this clarifies the contribution. Now to your comments about the biological findings:
>
> 1. [Sparsity]
> We agree that the fact that a model without L1 penalty performs worse does not mean that there is sparsity in the underlying data and we are open to suggestions on how to better substantiate that point. The unregularized model is clearly overfitting, and by using cross-validated regularization we can only get better. However, the fact that L1 regularization leads to such a strong improvement, does suggest that the sparsity assumption is pretty good. We would be grateful if you could expand on what you mean by “a more careful model selection analysis.”
>
> 2. [Novelty of center-surround/asymmetric RFs]
> You are right that these have of course been observed before (also long before Antolik 2016). However, what’s different is the prevalence of such non-Gabor RFs. For instance, if you look at Fig. 3 of Antolik 2016, you will notice that most RFs that are discernible are actually quite clean Gabors, while for the non-linear neurons (pink boxes in his Fig. 3) he does not obtain good RFs (unsurprisingly).
>
> 3. [Pathological RFs in Fig. 6]
> Thank you – this is a good suggestion. We will make sure to include such an analysis in the final version. Given our experience with activity maximization approaches, we expect them to look very similar overall, except for some of the noisy features in the surround (but certainly not more variable than across different neurons within the same group, which have different receptive field locations and have therefore seen different stimuli; see also response to AnonReviewer2).
>
> 4. [Should orientation be treated as a nuisance variable?]
> We agree that the visual system is neither equivariant to rotation nor to translation. However, that does not undermine the usefulness of equivariant representations at all. If your concern was valid and, for instance, horizontal filters looked completely different from vertical ones, then our rotation-equivariant network would not perform better than a regular CNN, because it would require a large number of features. But that’s not what we find. We find that the assumption of rotation equivariance (1) *does* lead to a substantial improvement over a regular CNN (see also our clarifications in the other comment) and (2) does so with fewer features, showing that it’s a good assumption to make. We added a sentence discussing this point (2nd sentence in the discussion). We will also add a plot showing the distribution of orientations being used at the readout stage for each feature, which will address the question of how rotation-equivariant the representation in mouse V1 actually is.
>
> 5. [Experimental details]
> We added those details (Section 4, subsection “Neural data”) and will show example traces in the final version.

---

> > ### Comment · AnonReviewer1 · 2018-11-19
> > **Mainly concerned about sparsity and RF structure**
> >
> > I acknowledge that I misunderstood the performance comparison. Providing a model with better fit quality is a considerable contribution.
> >
> > Regarding my other comments:
> >
> > 1. Sparsity: One obvious control would be to use L2 instead of L1 regularization. Are L2-regularized models statistically significantly worse than L1-regularized models? Or did I miss a reason for why trying L2 regularization is not possible?
> >
> > 2. Novelty of center-surround RFs: Perhaps this could be discussed in the text as you did in your response above.
> >
> > 3. RF structure: Thanks, I am looking forward to the new analyses. Just to add more detail, one reason why I am skeptical is that many of the RF features, e.g. in 12 and 13 in Figure 6, look very small. What is the size of these features (in degrees of visual space)? A scale bar in the RF plots would be helpful. How does this compare to published values for the resolution of the mouse visual system (e.g. both Niell and Stryker, 2008, and Marshel et al., 2011, report a mean preferred spatial frequency of about 0.04 cycles per degree)?
> >
> > 4. Orientation as a nuisance variable: Thanks for adding the additional discussion.
> >
> > 5. Thanks for adding additional details.

---

> > > ### Author Response · Authors · 2018-11-22
> > > **Good suggestions**
> > >
> > > Great we got the performance comparison sorted out!
> > >
> > > Thanks for your suggestions. We should have thought about using L2 for the feature weights and did this control now. Indeed, as you probably expected, the performance is now better than for the model without regularization of feature weights. However, the performance is still worse than using L1 regularization (L1: 0.47 vs. L2: 0.43; see updated Table 1). Thus, although the difference is not as extreme as our original (flawed) comparison suggested, but still substantial. We believe this control provides evidence that sparsity is a valid assumption.
> > >
> > > We also added scale bars to Fig. 6. The crops that are shown are 80x80 deg, i.e. covering three cycles for the *average* neuron. Classical RFs in mouse V1 can be as small as 5 deg (i.e. tiny in comparison to the crops shown in Fig. 6), which could explain why some of the features look rather small.

---

> > > > ### Comment · AnonReviewer1 · 2018-11-26
> > > > **Thanks for the additional controls**
> > > >
> > > > I am now more confident that there is some sparse structure in the RF weights. Regarding the RF structure, see my other comment.

---

> > > ### Author Response · Authors · 2018-11-26
> > > **Additional analyses of RF structure/robustness of preferred stimuli**
> > >
> > > We performed the control analysis you suggested, splitting the data into two halves, fitting models on each half and then computing the preferred stimuli (see new Fig. A.1). In addition, we also computed the preferred stimuli for another model, which used a different initialization and different hyperparameters, but all the data.
> > >
> > > The bottom line of the analysis is that the main finding holds robustly: Preferred stimuli are much more global than linearized (gradient) receptive fields. At the same time, it also shows that these preferred stimuli do show quite some idiosyncrasies – as you expected.
> > >
> > > However, unfortunately the analysis using the two halves of the data is not as telling as one would hope. The main issue is that with only half of the data available, the model fits are not as good and the preferred stimuli are not very reproducible (they often look really poor, indicating a poor fit). Using two different initializations, in contrast, produces more reproducible results.
> > >
> > > Overall, we agree that one should take these images with a grain of salt, but we do think they reveal an interesting and robust difference between the linearized RFs and the nonlinear function of the neurons. Due to time constraints we have not had the time to add a fully nuanced discussion of these issues, but will certainly make sure to not overstate anything in the final version. Any suggestions where we should improve the wording/claims would be very welcome.

---

> > > > ### Comment · AnonReviewer1 · 2018-11-26
> > > > **Would not use activity maximization RFs at this point**
> > > >
> > > > Thanks for performing the additional analyses. I appreciate your work doing these additional controls. Unfortunately, I think the new controls (Figure A.1) are textbook evidence for overfitting and confirm my concerns.
> > > >
> > > > The gradient receptive fields (row 1) seem to suggest that the additional structure in the RFs is not due to the rotation equivariance, but due to the activity maximization procedure. The split dataset experiments (row 4 and 5) strongly suggests that the activity maximization magnifies data-dependent differences in the models (i.e. magnifies over-fit features in the RF structures). I found hardly any matching structure in the two 50%-models, except for the classical RF structure that was already present in the Gradient RFs. Because this is likely due to differences in the training data, not in the weight initialization, it is unsurprising that models with different initializations are more similar.
> > > >
> > > > Even though models fit on 50% of the data perform worse, the RFs they produce are qualitatively similar to the "All data" model, suggesting that the 50%-models behave similarly. To me, this suggest that the activity maximization method for determining RFs is not robust enough to draw conclusions about RF structure from it at this point.
> > > >
> > > > If I were you, I would not trust this structure to be biologically meaningful. To show that the additional structure in the activity-maximization RFs is biologically meaningful despite the lack of consistency between datasets, it would be necessary to actually show these stimuli to an animal and test if they elicit stronger responses.
> > > >
> > > > My suggestion for the paper would be to remove the activity maximization and instead use the Gradient RFs. The contributions are then the improved model fit quality, and the result that L1-regularization improves performance over L2-regularization.

---

> > > > > ### Author Response · Authors · 2018-11-27
> > > > > **Removed activity maximization**
> > > > >
> > > > > Thanks for the quick feedback! We believe you have a point and therefore decided to remove the activity maximization figure from the paper and show only the linearized/gradient receptive fields.
> > > > >
> > > > > Although we do not entirely share your negative view on this analysis (we took great care not to overfit, using cross-validated regularization and early stopping), we do realize that burden of proof is on us and that at this point we do not have direct experimental evidence.
> > > > >
> > > > > Thanks again for the responsiveness and constructive feedback! It really improved the paper. We are now uploading a final revision and would appreciate if you update your score to reflect our discussions and the final version of the paper.

---

> > > > > ### Author Response · Authors · 2019-04-11
> > > > > **Verified experimentally that activity maximization produces meaningful images**
> > > > >
> > > > > > If I were you, I would not trust this structure to be biologically meaningful.
> > > > > > To show that the additional structure in the activity-maximization RFs is
> > > > > > biologically meaningful despite the lack of consistency between datasets,
> > > > > > it would be necessary to actually show these stimuli to an animal and test
> > > > > > if they elicit stronger responses.
> > > > >
> > > > > For the interested reader, we performed the experimental verification that activity maximization produces meaningful images in the meantime: https://www.biorxiv.org/content/early/2018/12/28/506956

---

### Official Review · AnonReviewer3 · 2018-11-03
**Interesting contribution to V1 modeling**

**Rating:** 7
**Confidence:** 4

**Review:**

In this interesting study, the authors show that incorporating rotation-equivariant filters  (i.e. enforcing weight sharing across filters with different orientations) in a CNN model of the visual system is a useful prior to predict responses in V1. After fitting this model to data, they find that the RFs of model V1 cells do not resemble the simple Gabor filters of textbooks, and they present other quantitative results about V1 receptive fields. The article is clearly written and the claims are supported by their analyses. It is the first time to my knowledge that a rotation-equivariant CNN is used to model V1 cells.

The article would benefit from the following clarifications:

1. The first paragraph of the introduction discusses functional cell types in V1, but the article does not seem to reach any new conclusion about the existence of well-defined clusters of functional cell types in V1. If this last statement is correct, I believe it is misleading to begin the article with considerations about functional cell types in V1. Please clarify.

2. For clarity, it would help the reader to mention in the abstract, introduction and/or methods that the CNN is trained on reproducing V1 neuron activations, not on an image classification task as in many other studies (Yamins 2014, etc).

3. “As a first step, we simply assume that each of the 16 features corresponds to one functional cell type and classify all neurons into one of these types based on their strongest feature weight.” and “The resulting preferred stimuli of each functional type are shown in Fig. 6.“
Again, I think these statements are misleading because they suggest that V1 cells indeed cluster in distinct functional cell types rather than form a continuum. However, from the data shown, it is unclear whether the V1 cells recorded form a continuum or distinct clusters. Unless this question is clarified and the authors show the existence of functionally distinct clusters in their data, they should preferably not mention "cell types" in the text.

Suggestions for improvement and questions (may not necessarily be addressed in this paper):

4. “we apply batch normalization”
What is the importance of batch normalization for successfully training the model? Do you think that a sort of batch normalization is implemented by the visual system?

5. “The second interesting aspect is that many of the resulting preferred stimuli do not look typical standard textbook V1 neurons which are Gabor filters. ”
OK but the analysis consists of iteratively ascending the gradient of activation of the neuron from an initial image. This cannot be compared directly to the linear approximation of the V1 filter that is computed experimentally from doing a spike-triggered average (STA) from white noise. A better comparison would be to do a single-step gradient ascent from a blank image. In this case, do the filters look like Gabors?

6. Did you find any evidence that individual V1 neurons are themselves invariant to a rotation?

7. The article could be more self-contained. There are a lot of references to Klindt et al. (2017) on which this work is based, but it would be nice to make the article understandable without having to read this other article.

Typo: Number of fearture maps in last layer

Conclusion:
I believe this work is significant and of interest for the rest of the community studying the visual system with deep networks, in particular because it finds an interesting prior for modeling V1 neurons, that can probably be extended to the rest of the visual system. However, it would benefit from the clarifications mentioned above.

---

> ### Author Response · Authors · 2018-11-13
> **Thoughtful review with good suggestion that we address**
>
> Thank you for your thoughtful and constructive review. Below we respond to your seven comments:
>
> 1. [Functional cell types]
> Finding out whether V1 is organized in distinct, well-defined clusters of functional cell types is indeed the big biological question we’re after. As you point out correctly, we do not answer this question in the present paper. The contribution of the present paper is not the biological finding that there are such well-defined cell types, but instead the development and verification of methods that allow us to address this question. With current methods (i.e. Klindt et al. 2017) we could not answer this question, because two Gabor filters with identical parameters except orientation would be considered two different cell types, which is undesirable from a biological perspective. The methods presented in our paper overcome this gap and let us treat preferred orientation as a nuisance just like receptive location. Therefore, we think it is appropriate to start the paper with these considerations, as they put our work in context by stating the long-term goals. We rephrased the introduction (third paragraph) to state the contributions more clearly. Please let us know if this revision addresses your concern or if you think that a more substantial revision of the introduction is necessary.
>
> 2. [CNN trained on neural data, not image categorization]
> Thank you. We revised the abstract (2nd sentence), introduction (third paragraph) and Section 3 (last sentence of first paragraph). Let us know if it’s still not clear.
>
> 3. [Functional cell types #2]
> You have a point. We changed the wording to “functional groups” wherever we describe what we did. The only place where we refer to functional cell types is the introduction where we provide the background/context of our work.
>
> 4. [Batch norm]
> Batch normalization serves two purposes here: (1) it helps training and (2) it ensures the features of the last CNN layer have unit variance, which is useful given the L1 penalty on the readout weights. Note that at test time, it does not have an effect. The normalization constants can be fully absorbed into the linear weights. In other words, if we gave you a trained model, you would not be able to tell whether it was trained with batch normalization or without, because they are indistinguishable at test time.
>
> 5. [Non-standard filters]
> We agree that they are not directly comparable. The point is that this model-based procedure reveals deviations from the linear models. It not only shows that spike-triggered average from white noise is insufficient for characterizing V1 neurons, but also provides a means for characterizing them. Regarding your question about a single gradient step from a blank image: these indeed tend to look more similar to standard Gabor filters (we performed such a comparison in a different project not using rotation equivariance and not published yet). We can create a new Figure analogous to Fig. 6 but using a single gradient step and add it to the final version of the paper.
>
> 6. [Rotation-invariant neurons]
> No, except for the trivial ones that have circularly symmetric center-surround receptive fields (or preferred stimuli), we did not find any evidence for rotation invariant neurons.

---

> > ### Comment · AnonReviewer3 · 2018-11-14
> > **A rigorous rebuttal**
> >
> > I would like to thank the authors for addressing thoroughly all my concerns. These clarifications confirm to me the rigor and quality of the work.
> >
> > "Regarding your question about a single gradient step from a blank image: these indeed tend to look more similar to standard Gabor filters (we performed such a comparison in a different project not using rotation equivariance and not published yet). We can create a new Figure analogous to Fig. 6 but using a single gradient step and add it to the final version of the paper."
> >
> > I think this would be a valuable addition to the paper because it shows how the model can also account for our old conception of V1 RFs (gabor-like RFs obtained from white noise stimulation, which is the filter corresponding to the best linear approximation of the cell's response). It might also help addressing the concern of Rev. 1: "Many of the receptive fields in Figure 6 look pathological (overfitted?) compared to typical V1 receptive fields in the literature."
> >
> > To justify the mathematical equivalence of a one-step gradient ascent from a blank image to STA in response to a perturbative white-noise stimulus, I think you could cite:
> >
> > Melinda E. Koelling and Duane Q. Nykamp. Computing linear approximations to nonlinear neu-
> > ronal response.Network (Bristol, England), 19(4):286–313, 2008. ISSN 1361-6536. doi:10.1080/09548980802503139
> >
> > Odelia Schwartz, Jonathan W. Pillow, Nicole C. Rust, and Eero P. Simoncelli. Spike-triggered neural
> > characterization.Journal of Vision, 6(4):13, July 2006. ISSN 1534-7362. doi: 10.1167/6.4.1

---

> > > ### Author Response · Authors · 2018-11-23
> > > **Gradient RFs added**
> > >
> > > We just uploaded a revision with a new Figure showing the gradient RFs (Fig. 7) of the same neurons as in Fig. 6.
> > >
> > > The gradient RFs mostly look like Gabor filters and are much more localized than the preferred stimuli. This result is indeed reassuring, as it shows that with similar visualization methods, we obtain similar results as previous work. Thanks for the suggestion!

---

> > > > ### Comment · AnonReviewer3 · 2018-11-24
> > > > **A useful addition**
> > > >
> > > > Looks good!
> > > >
> > > > I look forward to the result of the experiment proposed by Rev. 1, where you split the data in two halves, to see whether the preferred stimuli shown in fig.6 are robust to different fits of the model.

---

> > > > > ### Author Response · Authors · 2018-11-26
> > > > > **Done**
> > > > >
> > > > > We performed the analysis and added Figure A.1 to the most recent revision. Detailed response see https://openreview.net/forum?id=H1fU8iAqKX&noteId=BJgDxJhrC7&noteId=BylsquoYCQ

---

### Public Comment · (anonymous) · 2018-11-21
**Some questions on Fig. 6**

I'm curious about the preferred stimuli shown in Fig. 6.

1) Can you show a brief population summary on the preferred stimuli? e.g., what percentage of the cells exhibited Gabor filters? (cf., Fig 5D in [1])

2) If the preferred stimulus of many V1 neurons are not Gabor filters, why most previous researches failed to reveal these classes? For example, a recent paper [2] have fitted CNN on the image-response function of V1 neurons and performed activity maximization (I think the approach looks very similar to yours though it’s not cited in the "Related work" section), but in this paper, most preferred images are well fitted to Gabor (Fig 5E). I am interested in where the differences arise. Is it because of the high fitting accuracy of the rotation-equivariant CNN? Or just overfitting of the activity maximization process?

3) Also, if you presented drifting gratings during the two-photon imaging experiments, I'd like to know whether the neurons like cluster #12 or #13 are tuned to the orientations.

[1] Tang, S., Lee, T. S., Li, M., Zhang, Y., Xu, Y., Liu, F., ... & Jiang, H. (2018). Complex pattern selectivity in macaque primary visual cortex revealed by large-scale two-photon imaging. Current Biology, 28(1), 38-48.

[2] Ukita, J., Yoshida, T., & Ohki, K. (2018). Characterization of nonlinear receptive fields of visual neurons by convolutional neural network. bioRxiv, 348060.

---

> ### Author Response · Authors · 2018-11-22
> **Good questions, but non-trivial to address and beyond the scope of this paper**
>
> Thanks for your comments. As we also pointed out in the response to the reviewers, Fig. 6 is just a teaser and certainly should not be taken as a full-fledged analysis of non-linear response properties (which is a highly non-trivial problem in itself). We will try to make this point more explicit in the final version. A detailed analysis is forthcoming, but beyond the scope of this paper, which establishes the rotation-equivariant model as fitting the data better than regular CNNs.
>
> To briefly address your questions:
>
> 1) The preferred stimuli are obtained non-parametrically by directly optimizing in image space. What would be a good way of summarizing them? We could try to come up with some sort of classification, but we are not aware of a principled way of doing so. An interesting idea for future work would be to run the stimulus battery from [1] through our model and perform their classification in-silico.
>
> 2) Researchers have indeed found non-Gabor optimal stimuli (you're citing one of them [1]). With respect to [2], they did not systematically investigate deviations from Gabor filters. The only piece of evidence for Gabors in their study are two example neurons and the relatively high average correlation with a fitted Gabor (their Fig. 5E). However, note that a Gabor with a low carrier frequency is a Gaussian blob, which will generate a relatively high correlation with a center-surround kernel. Also, note that there are a couple of differences to our study:
>
>    a) Their optimization for finding optimal stimuli runs only for 10 iterations, which is probably closer to a linear approximation than running the optimization until convergence (as we do).
>
>    b) Their stimuli are masked with an aperture of 60 deg, whereas ours are 120x90 deg. Therefore, it is possible that our stimuli elicited stronger surround modulation (the optimal images shown in our Fig. 6 are 80x80 deg crops).
>
> 3) We did not present gratings in this experiment. However, there is evidence [3] that preferred orientations predicted from CNNs fit to natural image data generally match those obtained using oriented stimuli. Since #12 has pretty clearly oriented features in the center, we would indeed expect those cells to be tuned to orientation. #13 is less clear and would require more detailed investigation.
>
> [3] Sinz et al., bioRxiv. https://www.biorxiv.org/content/early/2018/10/25/452672

---

> ### Comment · AnonReviewer3 · 2018-11-22
> **Overfitting test**
>
> Thank you for your interesting comments.
>
> About your question: "Is it because of the high fitting accuracy of the rotation-equivariant CNN? Or just overfitting of the activity maximization process?"
>
> How would you distinguish between these two possibilities? Would the suggestion of Rev. 1 address your concern:
>
> "Perhaps a comparison of RFs learned on two disjoint subsets of the training set would help to determine which features are reproducible."

---

### Author Response · Authors · 2018-11-27
**Final revision uploaded**

We just uploaded a final revision addressing all reviewers' comments and ongoing discussions. Here are the main changes since the beginning of the rebuttal period:

- Removed language of "cell types" when referring to our analyses (R3).
- Performed proper control for claim that readout weights are sparse (R1).
- Replaced Fig. 6 (preferred stimuli) by linearized/gradient receptive fields to address R1's concern that preferred stimuli may be overfit.

We would like to thank all reviewers for their very constructive feedback and great responsiveness during the discussion period. It has really improved the paper. We would also like to ask all reviewers to make sure they review their scores and make sure they reflect the current version of the paper.

---

### Meta-Review · Area_Chair1 · 2018-12-15
**Consensus is accept**

**Confidence:** 5
**Recommendation:** Accept (Poster)

**Metareview:**

The overall consensus after an extended discussion of the paper is that this work should be accepted to ICLR. The back-and-forth between reviewers and authors was very productive, and resulted in substantial clarification of the work, and modification (trending positive) of the reviewer scores.